# Do Long COVID and COVID Vaccine Side Effects Share Pathophysiological Picture and Biochemical Pathways?

**DOI:** 10.3390/ijms26167879

**Published:** 2025-08-15

**Authors:** Jean-François Lesgards, Dominique Cerdan, Christian Perronne

**Affiliations:** 1Independent Researcher, 13397 Marseille, France; 2Independent Researcher, 33000 Bordeaux, France; 332017086@resopharma.fr; 3Infectious Diseases Department, University Hospital Raymond Poincaré, APHP, Université de Versailles Saint-Quentin-Paris Saclay, 92380 Garches, France; christianperronne@gmail.com

**Keywords:** long COVID pathophysiology, long COVID predictors, COVID vaccine side effects, immuno-inflammatory pathways, oxidative stress, complement pathway, renin angiotensin pathway, kininogen-kinin-kallikrein system

## Abstract

COVID affects around 400 million individuals today with a strong economic impact on the global economy. The list of long COVID symptoms is extremely broad because it is derived from neurological, cardiovascular, respiratory, immune, and renal dysfunctions and damages. We review here these pathophysiological manifestations and the predictors of this multi-organ pathology like the persistence of the virus, altered endothelial function, unrepaired tissue damage, immune dysregulation, and gut dysbiosis. We also discuss the similarities between long COVID and vaccine side effects together with possible common immuno-inflammatory pathways. Since the spike protein is present in SARS-CoV-2 (and its variants) but also produced by the COVID vaccines, its toxicity may also apply to all mRNA or adenoviral DNA vaccines as they are based on the production of a very similar spike protein to the virus. After COVID infection or vaccination, the spike protein can last for months in the body and may interact with ACE2 receptors and mannan-binding lectin (MBL)/mannan-binding lectin serine protease 2 (MASP-2), which are present almost everywhere in the organism. As a result, the spike protein may be able to trigger inflammation in a lot of organs and systems similar to COVID infection. We suggest that three immuno-inflammatory pathways are particularly key and responsible for long COVID and COVID vaccine side effects, as it has been shown for COVID, which may explain in large part their strong similarities: the renin–angiotensin–aldosterone system (RAAS), the kininogen–kinin–kallikrein system (KKS), and the lectin complement pathway. We propose that therapeutic studies should focus on these pathways to propose better cures for both long COVID as well as for COVID vaccine side effects.

## 1. Introduction

Long COVID (also termed “post-acute sequelae of COVID-19” or PASC) is a multi-systemic condition and often debilitating illness characterized by frequently severe symptoms that persist following infection with severe acute respiratory syndrome coronavirus 2 (SARS-CoV-2).

The worldwide burden of long COVID has been estimated to affect approximately 400 million individuals, with associated annual economic losses nearing 1 trillion USD, corresponding to nearly 1% of the global economy [1]. Notably, substantial variability exists in the reported prevalence of long COVID across different populations. Rigorous studies have evaluated that around 18 million US adults could experience long COVID [2], which is in line with the CDC estimation of 6% of the US population [3]. Also, estimates from the Institute for Health Metrics and Evaluation and the World Health Organization suggest that, during the initial three years of the COVID-19 pandemic, approximately 1 in 30 individuals in Europe (equating to around 36 million people) may have developed long COVID [4]. Data from extensive retrospective cohort analyses indicate that approximately 29.4% of individuals who had recovered from COVID-19 required rehospitalization, and 12.3% of these patients died after discharge, with most deaths attributed to post-COVID-19 complications [5].

Long COVID represents over 200 documented symptoms affecting the respiratory, cardiovascular, neurological, endocrine, urinary, and immune systems. Common symptoms include fatigue, dyspnea, persistent cough, anosmia, cognitive impairment, myocarditis, arrhythmias, and musculoskeletal pain [6]. Beside this, many of these manifestations also appear following COVID-19 vaccination, which has been described in a great number of publications and named post-acute-COVID-19 vaccination syndrome (PACVS) or, more simply, post-COVID-19 vaccination syndrome (PCVS) [6].

Additionally, emerging evidence indicates that both the SARS-CoV-2 virus and the SARS-CoV-2 spike protein may persist in the circulation or tissues for an extended period following infection and mRNA-based COVID-19 vaccination. This prolonged presence of the spike protein has raised concerns regarding its potential to sustain low-grade or chronic inflammatory responses in multiple organ systems, possibly contributing to adverse post-vaccination effects or long-term sequelae. In this context, we propose that three convergent immuno-inflammatory pathways that have been shown to be activated across severe COVID-19 may also be significant in long COVID and post-vaccinal syndromes and which may underlie overlapping clinical manifestations. These include (1) the renin–angiotensin–aldosterone system (RAAS); (2) the kallikrein–kinin system (KKS); and (3) the lectin complement pathway [7]. We propose the hypothesis that dysregulation of these interconnected cascades may play a central mechanistic role in the persistence and propagation of chronic inflammation and tissue damage observed in these two conditions.

## 2. Definition of Long COVID. Pathophysiological Manifestations, Predictors and Causes

### 2.1. Definition of Long COVID and Pathophysiological Manifestations

Long COVID is often categorized into distinct phases based on symptom duration: the “post-acute phase,” defined by the persistence of symptoms beyond three weeks following initial SARS-CoV-2 infection, and the “chronic phase,” characterized by symptoms extending beyond 12 weeks post-infection [8]. The symptoms of long COVID, like for acute COVID-19, result from damage to various organs and systems such as the respiratory, cardiovascular, neurological, endocrine, urinary, and immune systems. Several studies have documented multi-organ involvement in individuals with COVID-19. A prospective investigation conducted in individuals categorized as low-risk assessed six major organs, namely, the heart, lungs, liver, kidneys, pancreas, and spleen, and identified that 70% of the 201 participants exhibited dysfunction in at least one organ, whereas 29% showed involvement of multiple organs, indicating widespread multi-organ impairment following SARS-CoV-2 infection [9].

Long COVID has been linked to an array of pathological conditions affecting multiple systems, including cardiovascular, thrombotic, and cerebrovascular diseases [10]; metabolic disturbances such as type 2 diabetes [11]; and post-viral syndromes characterized by persistent fatigue, notably myalgic encephalomyelitis/chronic fatigue syndrome (ME/CFS) [12,13], dysautonomia, and heart dysfunctions, including postural orthostatic tachycardia syndrome [14]. To date, over 200 distinct symptoms have been documented, affecting multiple organ systems and reflecting the multisystemic nature of the condition [15]. The list of long COVID symptoms includes fatigue, headaches, dyspnea, sore throat, persistent cough, and smell loss; cardiac abnormalities such as myocarditis, microvascular angina, cardiac arrhythmias, and blood pressure abnormalities; cognitive and attention impairments and severe neurological alterations; sleep disturbances; and muscle pain (Table 1) [8,16,17,18]. Multi-system complaints were also reported as long COVID symptoms, including ongoing fever and gastroenterological symptoms [8]. A recent meta-analysis identified fatigue (58%), headache (44%), cognitive impairment (27%), alopecia (25%), and dyspnea (24%) as the five most commonly reported symptoms associated with long COVID [19].

The recovery from a mild SARS-CoV-2 infection usually takes 7–10 days from the onset of symptoms, but severe sickness recovery can take between 3 to 6 months. Numerous studies have projected that individuals recovering from COVID-19 following intensive care unit (ICU) admission may experience persistent symptoms, potentially enduring for several years [51]. Additional evidence indicates that individuals with long COVID are at an elevated risk of developing serious chronic conditions over time, as reflected by post-COVID-19 diagnoses of major adverse cardiovascular complications, chronic kidney and liver pathologies, and various respiratory disorders [52,53]. As mentioned, long COVID symptoms can last for years, and particularly in cases of new-onset ME/CFS and dysautonomia, which are expected to be lifelong [54]. Long COVID has been associated with substantial reductions in work capacity, with many affected individuals unable to return to employment, which is contributing to labor shortages and strongly affects the economy [15,55].

Until today there are currently no validated effective treatments against long COVID.

### 2.2. Predictors and Causes of Long COVID

Statistically significant predictors of long COVID have been proven to be significant: Arjun MC et al. (2022) have shown that several factors are independently associated with an increased risk of developing long COVID in 371 individuals with a median follow-up of 44 days. These include a greater number of symptoms during the acute infection phase (adjusted odds ratio [aOR] = 11.24; 95% confidence interval [CI]: 4.00–31.51), receipt of two doses of COVID-19 vaccine prior to infection (aOR = 2.32; 95% CI: 1.17–4.58), increased severity of the initial illness (aOR = 5.71; 95% CI: 3.00–10.89), and hospitalization during the acute phase (odds ratio [OR] = 3.89; 95% CI: 2.49–6.08) [56]. It is important to note here that receiving two doses of COVID-19 vaccine increased by 2.32 the risk of having long COVID, indicating a possible link between long COVID and vaccine side effects. Potential risk factors for long COVID also include female sex, type 2 diabetes, the presence of specific autoantibodies [57], connective tissue disorders [58], attention deficit disorders, chronic urticaria, and allergic rhinitis [23]. However, approximately one-third of individuals with long COVID do not present with any identifiable pre-existing conditions, although a third of people with long COVID have no identified pre-existing conditions [59].

In addition to these predictors, there are likely multiple potential overlapping causes which are upstream processes and biological drivers for long COVID. The following hypotheses for long COVID pathogenesis are proposed.

#### 2.2.1. Persistence of the Virus

Concerning neurologic issues and other damages linked with long COVID, the dysfunctions observed might be due to the persistence of the virus, which has not been completely eradicated, at the systemic, tissue, and/or cerebral levels [8,60,61,62,63]. Several studies have suggested that viral persistence may contribute to the pathogenesis of long COVID. SARS-CoV-2 RNA and/or viral proteins have been detected in multiple tissues and fluids of affected individuals, including the cardiovascular, nervous, and reproductive systems, as well as in skeletal muscle, ocular tissue, lymph nodes, appendix, breast tissue, lungs and liver tissues, plasma, stool, and urine [64,65]. Persistent circulation of the SARS-CoV-2 spike antigen has been documented in a substantial proportion of individuals experiencing long COVID, with 60% of a cohort of 37 patients exhibiting detectable levels up to one year following initial infection, whereas none of the 26 individuals who had recovered without post-acute sequelae (PASC) demonstrated detectable antigen [62]. Among 1569 biological samples collected from 706 individuals, 21% (95% CI, 18–24%) tested positive for at least one of the following viral antigens: S1 subunit, full-length spike, or nucleocapsid protein. Notably, detection of the spike protein peaked between 4 and 7 months following the initial SARS-CoV-2 infection (20%; 95% CI, 18–22%) [66]. Also, Tejerina F et al. (2022) found in a cohort study that 45% of COVID-19 patients with persistent symptoms had detectable plasma SARS-CoV-2 RNA [67].

Evidence from gastrointestinal tissue analyses has demonstrated the detection of SARS-CoV-2 components in some individuals, supporting the hypothesis that the gastrointestinal tract may serve as a site of viral persistence [68,69]. It is possible for RNA viruses to persist for months to years, as demonstrated with Ebola, Zika, and measles [70]. Fat cells and the endothelium may also be a reservoir [71,72]. But again, the most straightforward and accessible reservoir is likely to be in the gut, as residual viral nucleocapsid protein was identified in the intestinal epithelium of 24 out of 46 patients after 7 months [73] and 16 (30%) of 53 solid tissue samples collected at 1 month, 38 (27%) of 141 collected at 2 months, and 7 (11%) of 66 collected at 4 months, indicated viral presence [74].

#### 2.2.2. Altered Endothelial Function and Unrepaired Tissue Damage

Post-acute sequelae of COVID-19 (PASC) in the respiratory systems as well as in cardiovascular, nervous, immune, and other systems are also a field for long COVID to grow and worsen, especially in the absence of treatments during COVID episodes and after recovery [10,27].

The COVID-19 disease is a multisystem disease due in part to the vascular endothelium injury [75]. Indeed, lasting effects and long-term sequelae could persist after the infection and may be due to persistent endothelial dysfunction [63,76,77]. In a cohort of 80 individuals presenting with persistent post-acute symptoms and subsequently diagnosed with long COVID/PASC, pathological features included fibrin amyloid microclots and aberrant platelet activation profiles for all the subjects [78]. Findings from the PROLUN study (Patient-Related Outcomes and Lung Function After Hospitalization for COVID-19) revealed that nearly half of the post-COVID-19 cohort exhibited right ventricular mild impairment and two-fold left ventricular diastolic dysfunction versus controls, while arrhythmias persisted in 27% of participants (204 patients with COVID-19 and 204 controls) at the 3-month follow-up [79].

Persistent abnormalities in pulmonary function tests and chest computed tomography (CT) imaging have been frequently documented among individuals recovering from COVID-19 [80,81,82]. Moreover, Positron emission CT (PET-CT) imaging has revealed metabolic alterations in both pulmonary and cerebral tissues of individuals with long COVID, often correlating with radiographic indicators of lung fibrosis when compared to healthy control subjects [83,84,85,86]. Compared to healthy controls (*n* = 44), individuals with long COVID (*n* = 35) showed significant bilateral hypometabolism in brain regions including the olfactory and orbital gyri, right temporal structures (amygdala, hippocampus), thalamus, brainstem (pons/medulla), and cerebellum (*p* < 0.001 at voxel level; *p* < 0.05 FWE-corrected at cluster level) [83]. In a subset of long COVID patients (4 out of 13), CT scans revealed pulmonary abnormalities with mild [^18^F]FDG uptake. Additionally, PET imaging indicated reduced metabolic activity in the right parahippocampal gyrus and thalamus (voxel-level uncorrected *p* < 0.001) [85].

Regarding neurological damages, the role of prolonged neuroinflammation in the onset of symptoms has been hypothesized in many studies to involve microglia activation, autoimmunity, or local microthrombosis or mitochondrial dysfunction [87]. Neuropathological examinations from autopsy studies of individuals with long COVID, alongside investigations employing hamster models, have indicated evidence of ongoing neuroinflammatory responses, including sustained activation of microglial cells [88,89]. Complementary findings from neuroimaging studies, particularly magnetic resonance imaging (MRI), have demonstrated notable structural brain changes in long COVID patients [90]. These include significant increases in gray matter volume (GMV) across various regions, including the frontotemporal cortex, insula, basal ganglia, amygdala, hippocampus, and thalamus bilaterally. Additionally, alterations in GMV have been observed in limbic structures and secondary olfactory regions, suggesting persistent central nervous system involvement [90]. Also, dysfunctional signaling in the brainstem and/or vagus nerve are also key drivers of long COVID symptoms [25,63].

#### 2.2.3. Absence of Treatment During COVID Episode

In a large placebo-controlled study of metformin, ivermectin, and fluvoxamine, those who received metformin for 2 weeks were less likely than those receiving placebo to be diagnosed with long COVID [91]. By day 300 following infection, the incidence of long COVID was significantly lower in individuals treated with metformin (6.3%, 95% CI 4.2–8.2) compared to those receiving placebo (10.4%, 95% CI 7.8–12.9). Metformin use was associated with a 41% relative risk reduction in long COVID occurrence (hazard ratio [HR] 0.59, 95% CI 0.39–0.89; *p* = 0.012). Although there was no positive effect of ivermectin in this study, others studies and retrospective studies should be led on the subject of treatment and early treatments used during COVID crisis. This should be processed especially with molecules like ivermectin, which is known for its anti-inflammatory activity and which inhibits the production of key inflammatory players involved in COVID-19, in particular NF-κB, IL1, and IL-6 in vitro and in vivo [92,93]. In particular, ivermectin could inhibit the formation of NF-κB induced by the activation of lectin-like receptors (CLRs) activated by the N protein of the virus [94].

Taken together, these studies suggest that early interventions for COVID-19, in particular with anti-inflammatory and anticoagulant activities, could mitigate long COVID and support the pursuit of a robust preventive agenda, which is currently lacking.

#### 2.2.4. Autoimmunity and Immune Dysregulation

Autoimmunity due to molecular mimicry between pathogen and host proteins could be a contributive cause of long COVID [63]. Several independent investigations have reported an increased prevalence of circulating autoantibodies in individuals with long COVID [57]. These include autoantibodies directed against angiotensin-converting enzyme 2 (ACE2) [95], β2-adrenergic receptors, muscarinic M2 acetylcholine receptors, angiotensin II type 1 receptors (AT1R), and the Mas receptor involved in angiotensin-(1–7) signaling [96], suggesting possible autoimmune mechanisms contributing to PASC.

Immune dysregulation in long COVID has been characterized by distinct T cell perturbations, including increased markers of T cell exhaustion and diminished counts of CD4^+^ and CD8^+^ effector memory T cells [97,98]. Additionally, a persistent pro-inflammatory innate immune profile has been reported, marked by elevated frequencies of CD14^+^CD16^+^ intermediate monocytes and sustained activation of myeloid cells expressing CD38 and HLA-DR, lasting up to eight months following an initial mild-to-moderate SARS-CoV-2 infection compared to uninfected individuals [99]. Another investigation demonstrated that individuals with long COVID exhibited persistently increased proportions of both intermediate (CD14^+^CD16^+^) and non-classical (CD14^−^CD16^+^) monocyte subsets, with these alterations detectable as long as 15 months following acute SARS-CoV-2 infection [100].

Moreover, dysregulated monocyte and macrophage responses have been implicated in driving excessive inflammation and coagulation disturbances, potentially resulting in sustained tissue injury [48]. Ongoing or maladaptive activation of proinflammatory signaling pathways within tissue-resident macrophages is thought to contribute to the broad range of organ-specific manifestations observed in long COVID. Macrophages, which are integral to immune surveillance and maintenance of tissue homeostasis, are widely distributed across organs, including brain-resident microglia, hepatic Kupffer cells, pulmonary alveolar and interstitial macrophages, and adipose tissue macrophages [49]. Additionally, mast cells have been implicated as potential contributors to the pathophysiology of long COVID [15,45,46]. Immune dysregulation in long COVID may also involve reactivation of latent pathogens, notably herpesviruses such as Epstein–Barr virus (EBV) and human herpesvirus 6 (HHV-6), among others [63,97,98,101].

#### 2.2.5. Human Leukocyte Antigens (HLA) Variability and Dormant Virus Activation

Some authors attribute the persistence of viral antigens in long COVID to the lack or weak protection conferred by human leukocyte antigens (HLA) against SARS-CoV-2 in individuals carrying HLA alleles with low binding affinities to the virus [66,102]. There is evidence of an association between SARS-CoV-2 antigen positivity and PASC related to several symptom domains. These authors have proposed that comprehensive characterization of HLA Class I and II alleles in individuals with long COVID could offer critical insights into the mechanisms of persistent antigen presence and its role in long COVID pathogenesis, potentially guiding future therapeutic approaches. Moreover, specific HLA variants, including HLA-DRB108* and HLA-DRB104*, have been linked to heightened disease severity and increased mortality risk in COVID-19 [103]. Various articles evoke that the HLA system and the COVID-19 outcome could be ethnic-dependent with some alleles being common between some populations, such as HLA-DRB1*15 and HLA-A*30:02 [103,104].

Moreover, Epstein–Barr virus (EBV) reactivation, detectable in the oropharyngeal region, has been observed at a higher frequency among individuals experiencing long COVID-related fatigue, even several months following acute SARS-CoV-2 infection, compared with recovered individuals without persistent symptoms [105,106]. This suggests that EBV replication may be a co-factor in a sub-group of patients developing long COVID fatigue. These findings suggest that many long COVID symptoms may not be a direct result of the SARS-CoV-2 virus but may be the result of COVID-19 inflammation-induced EBV reactivation [107]. Overall, these findings suggest differential effects of chronic viral coinfections on the likelihood of developing long COVID and association with distinct syndromic patterns [108], and it may be linked with HLA factors [105].

#### 2.2.6. Impacts of SARS-CoV-2 on the Microbiota

SARS-CoV-2 has a profound effect on the microbiota, including the virome [63,109,110,111]. Accumulating data indicate that individuals with long COVID frequently experience gastrointestinal symptoms, including abdominal discomfort, alongside persistent alterations in gut microbial composition (dysbiosis) [112]. In certain cases, sustained gut microbiota imbalances have been implicated in the exacerbation or maintenance of diverse long COVID manifestations, particularly those involving fatigue; musculoskeletal pain; gastrointestinal disturbances; and neuropsychiatric symptoms such as depression, anxiety, and headache [112]. The dysbiotic microbiome in long COVID patients has been shown to be associated with a decrease in beneficial short chain fatty acid-producing bacteria (*Faecalibacterium*, *Ruminococcus*, *Dorea*, and *Bifidobacterium*) and an increase in opportunistic bacteria (*Corynebacterium*, *Streptococcus*, *Enterococcus*) [113]. The gastrointestinal tract has not just a digestive function but also is responsible for achieving immune system homeostasis. Alterations in gut microbial communities (dysbiosis) can modulate systemic physiology by impacting immune function and contributing to pathological processes in distant organs through established pathways such as the gut–lung and gut–brain axes. Additionally, microbial-derived metabolites and microRNAs (miRNAs) are thought to mediate inter-organ communication, potentially influencing pulmonary, neurological, and systemic outcomes in the context of long COVID. Yeoh et al. pointed out that dysbiosis seen in COVID-19 patients drives inflammation and fuels long-term symptoms [109].

#### 2.2.7. Integration of Reverse-Transcribed SARS-CoV-2 RNA in the Human Genome

Several studies have reported the possibility that the viral genome could be retro-transcribed into DNA copies of SARS-CoV-2 sequences and integrated into human DNA in infected cells, thus becoming a driver and source of the synthesis of RNA and antigens of viral origin [114,115]. The authors showed that it was consistent with a LINE1 retrotransposon-mediated, target-primed reverse transcription and retroposition mechanism which may also happen after mRNA COVID vaccination [115]. Consequently, this would explain the persistence/recurrence of cardiovascular and neurological complications following SARS-CoV-2 infection representing part of the symptoms of long COVID, but this hypothesis is considered as very unlikely and would require an analysis of the human genome in patients, which has not been done yet.

## 3. Long COVID and Vaccine Side Effects: A Focus on the Common Cardiovascular and Neurological Damages and Associated Pathologies

### 3.1. Long COVID, Acute and Post-Acute COVID Vaccine Syndromes 

Some authors have proposed to refer to long COVID as post-acute COVID-19 syndrome (PACS) [6]. Post-acute COVID-19 syndrome (PACS) represents a heterogeneous condition characterized by a multifactorial etiology, involving diverse and potentially overlapping pathophysiological mechanisms. Distinct PACS subtypes may be delineated based on symptom profiles, severity, and temporal onset patterns [6]. It is also recognized that post-COVID-19 vaccination syndrome (PCVS) exists as shown by thousands of publications today on COVID vaccine side effects [7,116,117,118]. It is essential to differentiate between immediate adverse events following COVID-19 vaccination, termed acute COVID-19 vaccination syndrome (ACVS), which include rapid-onset hypersensitivity reactions such as anaphylaxis and allergic responses occurring shortly after mRNA vaccine administration [6], and delayed-onset conditions classified as post-acute COVID-19 vaccination syndrome (PACVS), which may manifest days to months following either mRNA or adenoviral vector-based COVID-19 immunization [6]. Authors have reported that 3–5% of the vaccine recipients experienced side effect symptoms for longer than 1 week, and 0.2–1.4% for longer than 1 month [119,120].

Fatigue is considered a non-severe adverse event symptom [8,20]. However, ACVS can also include anaphylaxis, anaphylactic reactions, anaphylactoid reaction, anaphylactic shock, and anaphylactoid shock [121,122] as well as vasovagal syncope/presyncope [123] that can follow immediately after vaccination and in the worst case can lead to death [124]. In this review, we are using the general term post-COVID-19 vaccination syndrome (PCVS) as we focus on post-acute COVID-19 vaccination syndrome (PACVS), i.e., chronic manifestations only.

### 3.2. Vascular Damages and Pathologies in Long COVID and After COVID Vaccination

#### 3.2.1. Introduction on Cardiovascular Damages

In cases of severe COVID-19, disruptions within the circulatory system have been characterized by endothelial injury, alongside elevated risks of thromboembolic complications such as deep vein thrombosis, pulmonary embolism, and hemorrhagic events [15,76,77,125]. Dysregulation of vascular function and coagulation pathways may have systemic consequences, potentially serving as a common mechanistic link underlying the wide range of long COVID manifestations across multiple organ systems [39]. Among the diverse clinical manifestations of long COVID, vascular and cardiac complications are frequently reported, including persistent endothelial injury; thrombotic events; and a range of cardiac dysfunctions such as myocardial inflammation (myocarditis), coronary microvascular dysregulation (microvascular angina), arrhythmogenic disturbances, and impaired blood pressure homeostasis (Table 1) [8,16,17,39].

Post-COVID-19 vaccination syndrome (PCVS) and COVID vaccine side effects also include very similar endothelial and cardiovascular damages to the endothelial wall, cardiac abnormalities such as myocarditis, and coagulation issues like thrombosis (Table 1) [6,7,118].

#### 3.2.2. Myocarditis in Long COVID and After COVID Vaccination

Myocarditis has been shown to be a long-term complication of COVID-19, and this inflammatory condition involving the myocardium can be fatal in the long term in cases of progression to ventricular dysfunction and heart failure, so long COVID patients must be followed carefully [32,126]. Blagova et al. (2022) documented a case series involving 14 individuals who developed myocarditis following recovery from acute COVID-19. In these cases, myocardial inflammation was confirmed by endomyocardial biopsy, with diagnoses established between 2 and 18 months after the initial infection [126]. Other important studies have found signs of inflammatory myopericardial involvement using cardiac magnetic resonance (CMR) imaging (myocardial mapping and scar imaging) techniques to detect diffuse inflammatory myocardial, which persisted several months after the initial COVID illness [34]. At a median baseline of 109 days, 73% of participants reported cardiac symptoms (myocardial and pericardial inflammation), and 53% of the cohort had persistent cardiac symptoms for the follow-up at 329 days after diagnosis of COVID-19 infection. The authors show that female gender as well as diffuse myocardial involvement on baseline imaging independently predicted the presence of cardiac symptoms at follow-up at 329 days [34].

Regarding COVID-19 vaccines at present, more than three years after licensing of these products, it has been established that the highest risk of developing myocarditis occurs in males aged 12–30 years, within 1–14 days post vaccination after the second dose of vaccination with an mRNA vaccine [33] or even between 1 and 7 days [35,127,128,129]. Presentation within 4 days is common, but there are also patients presenting 2 weeks or more after vaccination [130] and up to several months. A study by Gautam et al. (2021) reported a case of an elderly man with acute myocarditis on cardiac magnetic resonance imaging (CMRI), which was attributed to COVID-19 vaccine in the absence of any predisposing conditions or alternative risk factors, 3 months after receiving the second dose of mRNA vaccine [131]. Spike protein (but not the nucleocapsid protein) could be detected precisely within the foci of inflammation in both the brain and the heart, particularly in the endothelial cells of small blood vessels, in an individual that collapsed 2 weeks after the third dose of the COVID-19 vaccine and died 1 week after, which suggests an implication of vaccine-induced spike proteins rather than the virus itself [132]. More studies reporting the presence of both N and S proteins are needed in this matter in order to understand the consequences of long COVID vs. vaccine side effects. Vaccine-induced spike protein was also detected in cardiac tissue in individuals experiencing intramyocardial inflammation after COVID-19 vaccination, including a case with symptoms 21 days after vaccination and successful mRNA detection [133].

The vaccine dosing interval can also influence the risk of developing myocarditis post vaccination. In this regard, a study from Canada demonstrated that the extended interval of 8 weeks versus a 3–4 weeks interval between mRNA vaccine doses 1 and 2 was associated with a reduced risk of myocarditis and pericarditis particularly among male individuals aged 18–24 years [134]. Data indicate that the incidence of myocarditis following administration of the BNT162b2 (Pfizer) vaccine increases substantially when the dosing interval is shortened. Specifically, individuals receiving their second dose within 30 days or less exhibit a markedly elevated myocarditis rate with 52.1 cases per 1,000,000 doses (95% CI, 31.8–80.5), compared with those with an interval exceeding 56 days, where the rate drops to 9.6 cases per 1,000,000 doses (95% CI, 6.5–13.6), representing approximately a 5.4-fold increase. For the mRNA-1273 (Moderna) vaccine, a significantly higher incidence of myocarditis has been observed when the dosing interval between vaccine doses is shortened. Specifically, the myocarditis rate reached 83.9 cases per 1,000,000 doses (95% CI, 47.0–138.4) when the interval between doses was 30 days or less, compared to a substantially lower rate of 16.2 cases per 1,000,000 doses (95% CI, 10.2–24.6) when the interval exceeded 56 days, corresponding to an approximately 5.2-fold increase in risk [134]. In consequence to these observations, some countries like Australia, Canada, or the UK decided to extend the interval to 8–12 weeks between primary course doses, specifically for the very highest risk group which is around 12–17 years old [33].

Thus, it is clear that both long COVID and COVID vaccines can induce myocarditis. During the pandemic, health authorities and many physicians and researchers pushed for COVID vaccination even in patients with a prior history of acute myocarditis and offered reassurance based on the fact that it was not associated with a risk of myocarditis recurrence or serious side effects [135]. Regarding the large literature on myocarditis and severe cardiac issues induced by COVID vaccines, this appears to be debatable.

#### 3.2.3. Thrombosis in Long COVID and After COVID Vaccination

Thrombosis is another severe and worrying side effect in association with COVID-19, long COVID, and after COVID vaccination [15,40,136,137,138,139,140,141].

Endothelial inflammation, also called endothelitis or endothelialitis, is coupled with thrombotic processes which are observed during COVID-19, such as microthrombus formation, macrovascular thrombosis, and pulmonary embolism [142,143,144]. Pretorius et al. revealed common clotting pathologies in plasma of acute and long COVID patients, further supporting the existence of persistent microthrombi [78]. Also, long-term changes to the size and stiffness of blood cells have also been found in long COVID with the potential to affect oxygen delivery [145]. Moreover, persistent microvascular alterations, characterized by a sustained decrease in capillary density, have been observed in individuals with long COVID up to 18 months following acute infection in comparison to non-affected controls [146].

Like pericarditis and myocarditis, thrombosis following COVID vaccination has been shown to occur and was associated with strong levels of D-dimers and CRP [36,147,148]. Furthermore, thromboses were observed mainly in young subjects (22 to 49 years of age) after AstraZeneca vaccination, whereas severe forms of COVID-19 occur mainly in much older people and the elderly [147]. In this research, the participants experienced venous thromboses, comprising nine individuals diagnosed with cerebral thrombosis and three with pulmonary embolism, and six out of the eleven patients did not survive.

The presence of persistent fibrinolysis-resistant microthrombi has been documented in individuals during both the acute phase of COVID-19 and in long COVID, potentially playing a key role in the development of thrombotic complications, and should be part of diagnosis, including for vaccine side effects and the search of therapeutic targets [149].

Finally, for some authors, the activation of coagulation in severe forms is comparable to a picture of disseminated intravascular coagulation (DIC) with thrombocytopenia, associated with the consumption of coagulation proteins and elevation of D-dimers, and could be linked to the production and accumulation of platelet activation immune complexes [47]. The detoxification of these immune complexes normally carried out by the complement system could be compromised due to the consumption of complement factors throughout the immuno-inflammatory phase, which will be described in this review. In long COVID, the formation of immune complexes may lead to severe disease progression, highlighting the urgency for early detection and intervention [150]. Regarding post COVID vaccine issues, studies have identified that vaccine-induced immune thrombotic thrombocytopenia (VITT), which typically manifests 7 to 10 days following the initial administration of the ChAdOx1 nCoV-19 adenoviral vector vaccine, involves the formation of pathogenic immune complexes. These complexes appear to activate multiple innate immune mechanisms, promoting both platelet aggregation and leukocyte activation, thereby contributing to the thrombotic and thrombocytopenic manifestations observed in affected individuals [47].

### 3.3. Neurological Damages and Consequences on Cognitive Systems in Long COVID and After COVID Vaccination

Besides myocarditis/pericarditis due to SARS-CoV-2 infection/COVID-19, long COVID, and PCVS, several other severe health conditions are observed, including neurological damages (Table 1).

SARS-CoV-2 has a strong tropism towards the nervous system and in particular towards the central nervous system (CNS), which is a key and worrying target in COVID-19 as well as in long COVID [7,151,152,153]. Various studies indicate that the virus can traverse the olfactory or vagus nerve and penetrate the blood–brain barrier (BBB) [151,154,155]. Additionally, it can access the central nervous system via the blood–cerebrospinal fluid (CSF) barrier, which is more permeable than the BBB and consists of a single layer of endothelial cells from the choroid plexus [156,157]. Consequently, neurological and neurocognitive manifestations are frequently reported in long COVID, encompassing a broad spectrum of symptoms such as deficits in memory and executive function, sensory disturbances, paresthesia, vertigo, gait instability, photophobia, phonophobia, olfactory and gustatory dysfunction (including anosmia, hyposmia, or phantosmia), and features of autonomic nervous system dysregulation, all of which can significantly impair daily functioning [16,158].

When the virus is present in the brain, inflammation is triggered through the interaction between the spike protein and ACE2 receptors, which are expressed in the human CNS, particularly in the spinal cord, spinal ganglion, brainstem nigra, choroid plexuses, hypothalamus, hippocampus, middle temporal gyrus, and posterior cingulate cortex [154,159]. Evidence from both human studies and murine models of long COVID has demonstrated widespread cellular dysregulation across multiple immune and neural lineages, persisting even after mild infections. These alterations include pronounced microglial activation, analogous to neuroinflammatory processes observed in post-chemotherapy cognitive impairment (“chemo-brain” or “chemo-fog”), accompanied by disrupted hippocampal neurogenesis, reductions in oligodendrocyte populations, and significant myelin degradation [160]. Given that activation of hippocampal microglia has been implicated in virus-associated cognitive dysfunction [161], this mechanism could potentially underlie the neuropsychiatric manifestations observed in certain individuals following SARS-CoV-2 infection, including impairments in memory, excessive daytime sleepiness, persistent fatigue, and disturbances in sleep regulation such as insomnia [162].

Moreover, both the SARS-CoV-2 spike protein and the one introduced through vaccination can reduce serotonin levels by interacting with ACE2, potentially contributing to or worsening depression and even suicidal tendencies [163]. Long COVID has been associated with a range of audiovestibular disturbances, encompassing symptoms such as persistent tinnitus, varying degrees of hearing impairment, and episodes of vertigo [16,164]. Also, neuronal atrophy and degeneration of cranial nerves, including the olfactory nerve and the neighboring olfactory bulb, have previously been reported in patients with persistent hyposmia or anosmia after acute COVID-19 [8]. In a cohort of 23 patients with persistent anosmia due to long COVID, neuroimaging revealed structural alterations in the olfactory system. Specifically, olfactory cleft opacification, particularly in the mid and posterior segments, was observed in 73.9% of cases. Moreover, 43.5% exhibited reduced olfactory bulb volumes, and 60.9% showed shallowing of the olfactory sulci, consistent with COVID-19-associated olfactory neurodegeneration [165].

In agreement with these findings, patients with long COVID exhibited lower metabolic activity in their brain [158]. Vagus nerve stimulation mediating anti-inflammatory responses highlights the relationship between SARS-CoV-2 infection and the parasympathetic nervous system (PNS) and suggests the autonomic nervous system (ANS) as a therapeutic target. Of note, vagus nerve dysfunction has also been reported in SARS-CoV-2 infection and proposed as a key pathophysiological hallmark of long COVID [63].

Also, signs of peripheral nerve and muscular system dysfunction were also reported after SARS-CoV-2 infection. Persistent manifestations such as muscle pain, reduced physical endurance or exercise intolerance, abnormal sensory perceptions including paresthesia and neuropathic pain, as well as autonomic nervous system dysfunction have been consistently associated with long COVID in affected individuals [166].

After receiving COVID-19 vaccines, a significant number of healthy individuals develop PCVS with a variety of neurological complications and long-lasting neuropsychiatric symptoms similar to those experienced by a large proportion of COVID-19 survivors [22,167,168,169]. As with the SARS-CoV-2 virus, it has been demonstrated that both COVID vaccine mRNA and free spike proteins could enter the brain via the bloodstream and by penetrating the BBB [170,171]. Bahl et al. had already shown that the mRNA/LNP platform could reach the brain, as also specified by the European Medicines Agency (EMA) for up to 2–4% of the mRNA plasma concentration [172,173,174].

Neurological complications post-COVID vaccination include fatigue; headache; immune encephalitis; cerebral sinus venous thrombosis (VST); transverse myelitis (TM); Guillain–Barré syndrome and optic neuritis; ischemic stroke; intracerebral bleeding; hypophysitis; epilepsy; hyperactive encephalopathy; and acute, disseminated encephalomyelitis [21,22]. Transverse myelitis (TM) has been documented after COVID vaccination [21,175,176,177,178] as well as in COVID-19 and long COVID [179,180,181,182]. A review encompassing 28 studies identified 31 cases of post-COVID-19 vaccination-associated TM involving 17 females and 14 males with a mean age of 52 ± 19 years. Most cases followed the first vaccine dose (*n* = 24), and the Oxford-AstraZeneca (ChAdOx1 nCoV-19) vaccine was most frequently implicated (12 cases), followed by Pfizer [8], Moderna [7], Sinopharm [3], and Janssen [1,6]. Clinical outcomes were favorable (Modified Rankin Score < 3) in 21 patients. Notably, the incidence of post-vaccination TM appears considerably lower than that reported after SARS-CoV-2 infection, which is approximately 0.5 cases per million per year. Interestingly, a comparative analysis of 30 articles detailing 65 unique cases were included, of which 48 (73.8%) were infection TM and 17 (26.2%) revealed no substantial differences in latency to symptom onset, clinical or imaging characteristics, or recovery profiles between TM following SARS-CoV-2 infection and that occurring after COVID-19 vaccination, implying a potentially shared pathogenic mechanism [182]. Notably, about 80% of affected individuals experienced favorable neurological outcomes.

Venous sinus thrombosis (VST) represents a neurological complication associated with SARS-CoV-2 vaccination [183]. It is identified as the third most common adverse event related to vaccination, primarily attributed to vaccine-induced hypercoagulable states [22,183]. The underlying mechanisms proposed include direct platelet activation or indirect endothelial activation, both promoting a pro-thrombotic state [184].

Neurological adverse events involving the central and peripheral nervous systems (CNS and PNS) have been reported more frequently than complications affecting the cardiovascular, endocrine, renal, or dermatological systems [22]. Within the peripheral nervous system, skeletal muscle involvement has been observed in some cases [185], with manifestations such as myositis [186], dermatomyositis, immune-mediated necrotizing myopathy [185], rhabdomyolysis, and polymyalgia rheumatica.

Emerging evidence suggests that individuals with long COVID exhibit molecular signatures resembling those observed in Alzheimer’s disease [187], along with the presence of neurotoxic amyloidogenic peptides capable of forming insoluble aggregates [188]. In addition, widespread neuroinflammatory responses have been documented [189], together with region-specific reductions in cerebral and brainstem metabolic activity that correlate with distinct clinical manifestations [83,190]. Furthermore, cerebrospinal fluid abnormalities have been detected even in non-hospitalized long COVID patients, with younger individuals showing a tendency toward delayed onset of neurological symptoms [191].

To summarize, possible mechanisms for the neuropathologies present in both long COVID and PCVS include neuroinflammation with microglia activation, autoimmunity, damage to blood vessels induced by coagulopathy and endothelial dysfunction, and neuron injury [7,87,158,192]. The potential link between neurodegenerative diseases and long COVID, as well as the long-term effects of COVID-19 vaccination, warrants close monitoring in the global population [168].

### 3.4. Accelerated Biological Aging and Senescence in Long COVID and After COVID Vaccination

Telomere length in peripheral blood lymphocytes of COVID-19 patients aged 29 to 85 years has been found to be decreased in association with increased disease severity [193]. This could be linked with spike-induced aging as shown in a study which measured the biological age of 117 COVID-19 survivors and 144 uninfected volunteers. The authors observed that patients who had COVID-19 showed a marked increase in biological age averaging 10.45 ± 7.29 years (5.25 years above the range of normality) compared to an average increase of 3.68 years (±8.17 years) in uninfected individuals [194].

We suggested that this phenomenon may arise not only shortly after COVID-19 infection, particularly in cases where early treatment was not administered, but also in long COVID, and potentially following COVID-19 vaccination, especially after multiple doses, and that it is associated with inflammatory and oxidative stress-induced shortening of telomeres [7,195]. Besides, many of the symptoms of long COVID are typically seen with advancing age [26]. Our hypothesis for long COVID has been shown in a study performed on a large cohort of 3335 who found lower cognitive scores in individuals with more than 12 weeks of COVID-19 symptoms. These subjects were self-defined as having “long COVID” and met NICE “Post-COVID-19 syndrome” and WHO “Post COVID-19 condition” definitions [196,197]. These cognitive deficits were detectable nearly two years post-infection and were comparable to the effect of “an increase in age of approximately 10 years, or exhibiting mild or moderate symptoms of psychological distress”.

## 4. Possible Common Immuno-Inflammatory Patterns and Biochemical Aspects of Severe COVID, Long COVID, and COVID Vaccine Side Effects and Therapeutic Targets

### 4.1. Spike Protein: The Common Toxicant

In COVID-19, hyperinflammation and coagulopathy are triggered by both N (nucleocapsid) and S (spike) proteins [7,198,199].

Meanwhile, the spike protein alone activates various key inflammatory pathways which we have proposed may be common in COVID, long COVID, and PCVS [6,7,200]. A critical factor contributing to spike protein-associated toxicity lies in the broad distribution of ACE2 receptors across multiple tissues and fluids. ACE2 is expressed in numerous anatomical sites, including the upper and lower respiratory tract (pharynx, trachea, lungs), cardiovascular system (heart, vasculature, bloodstream), central nervous system (brain, cerebrospinal fluid), gastrointestinal tract, kidneys, and male reproductive organs. Moreover, ACE2 is detectable in various body fluids, such as mucus, saliva, urine, semen, and breast milk (Figure 1) [201]. Consequently, the spike protein can trigger inflammation across multiple organs and systems. This helps explain why most COVID-19 patients experience a range of symptoms beyond respiratory issues, including cardiovascular, neurological, gastrointestinal, and renal complications [28,202,203,204], as well as in people with long COVID and who have received COVID vaccines [6].

As already mentioned, long COVID-associated pathologies and damages can be linked to the persistence of the virus, which has not been completely eradicated, at the systemic, tissue, and/or cerebral levels [8,60,61,62,63], as viral proteins and/or RNA have been found in the reproductive system, cardiovascular system, brain, muscles, eyes, lymph nodes, appendix, breast tissue, hepatic tissue, lung tissue, plasma, stool, and urine in long COVID patients [64,65,67,68,205,206]. SARS-CoV-2 spike antigen has been detected in the bloodstream of 60% of individuals with long COVID up to one year post-infection, whereas it was undetectable among SARS-CoV-2-infected individuals without persistent symptoms [66]. As already mentioned, Tejerina F et al. (2022) found in a cohort study that 45% of COVID-19 patients with persistent symptoms had detectable plasma SARS-CoV-2 RNA [64].

However, these studies also show that long COVID symptoms in association with chronic inflammation may occur in the absence of SARS-CoV-2.

Given that the spike protein is a structural component of SARS-CoV-2 (including its variants) and is also synthesized following administration of mRNA and adenoviral vector-based vaccines, any potential spike protein-associated toxicity is relevant to both natural infection and these vaccine platforms, as they are based on the production of a very similar spike protein to the virus. Indeed, very importantly, Pfizer confirmed in a report, approved 26 December 2020, “We demonstrate that the BNT162b2 RNA sequence encodes a recombinant P2 S that can authentically present the ACE2 binding site and other epitopes targeted by SARS-CoV-2 neutralizing antibodies.” [207]. It was also shown in a work published in *Nature* with several authors from BioNTech and Pfizer companies that “Our binding and structural analyses indicate that the BNT162b2 RNA sequence encodes a recombinant S(P2) that can authentically present the ACE2-binding site and other epitopes that are targeted by SARS-CoV-2-neutralizing antibodies… The binding of expressed and purified S(P2) to ACE2 and a neutralizing monoclonal antibody further demonstrates the conformational and antigenic integrity of this prefusion-stabilized S” [208]. In consequence, the spike protein produced by these COVID vaccines may be able to interact with AE2 receptors and to trigger the associated biochemical pathways.

Lei et al. (2021) demonstrated that exposure to a pseudovirus expressing the SARS-CoV-2 spike (S) protein, specifically the S1 subunit containing the receptor-binding domain (RBD) but lacking viral RNA, induced inflammation and tissue damage in the lungs and arteries of mice following intratracheal administration [209]. Similar effects were observed in human epithelial cells, where the spike protein alone was shown to disrupt mitochondrial function, indicating that the spike protein itself is sufficient to induce the cardiovascular injuries observed in COVID-19. In the context of vaccination, beyond the presence of free circulating spike protein, its expression on endothelial cells may further contribute to vascular complications by activating platelets and promoting coagulation, potentially resulting in thrombosis through the release of platelet factor 4 (PF4) [147,210].

Furthermore, studies have demonstrated that both mRNA and free spike proteins can enter the brain via systemic circulation and by crossing the blood–brain barrier (BBB), raising concerns about potential inflammatory responses in cerebral vasculature and neural tissue [170,171]. Bahl et al. provided evidence that vaccine-derived mRNA can localize to the brain, a finding also acknowledged in reports by the European Medicines Agency (EMA) for up to 2% of the plasmatic mRNA [172,173,174]. Of note, anosmia has been described following vaccination in COVID-19-negative subjects. This shows that symptoms identical to COVID-19 can appear and have been triggered by the spike protein alone [211]. In the context of COVID-19, long COVID, and vaccine-associated adverse events, the spike protein has been implicated in triggering central nervous system (CNS) inflammation, primarily through the activation of microglia and astrocytes, which may represent a key mechanism underlying neurological toxicity [7]. Experimental studies have demonstrated that the S1 subunit of the spike protein can induce neuroinflammation in microglia via activation of the NF-κB and p38 MAPK signaling pathways, leading to elevated production of pro-inflammatory mediators such as TNF-α, IL-6, IL-1β, and nitric oxide (NO) [212].

Moreover, the spike protein has been detected several weeks following COVID-19 vaccination, raising concerns about its potential to induce prolonged or chronic inflammation across various organs [201,213,214]. Röltgen et al. demonstrated that mRNA vaccination induces persistent germinal center reactions, with vaccine-derived mRNA and spike antigens detectable in lymphoid tissues for up to eight weeks post-vaccination in some individuals [214]. Additionally, the concentration of free spike proteins in circulation after vaccination has been reported to reach levels comparable to those observed during SARS-CoV-2 infection (up to 150 pg/mL), potentially activating ACE2 receptors, particularly in tissues with high spike protein accumulation (Figure 1) [215]. These findings suggest that vaccination could, in some cases, elicit symptoms similar to those seen in COVID-19 and long COVID, and may theoretically contribute to the development or exacerbation of inflammatory conditions over the medium to long term, including cardiovascular, neurological, oncological, and autoimmune disorders. Cosentino and Marino (2022) pointed out that adverse effects of the COVID-19 vaccines could be related to excess SARS-CoV-2 spike production in specific individuals “for too long and/or in inappropriate tissues and organs”, while the probability of this occurrence “is at present unpredictable, since systemic biodistribution and disposition of the COVID-19 mRNA vaccine has so far never been considered an issue, and as a consequence it has never been studied as it would have actually deserved.” [215]. Indeed, Ota N et al. (2025) provided novel findings demonstrating the presence of mRNA vaccine-derived spike protein in the cerebral vasculature of 43.8% of individuals who experienced hemorrhagic stroke, detected as long as 17 months following vaccination, in the absence of documented SARS-CoV-2 infection [44].

Regarding the importance of spike proteins lasting in the body and their long-term effects even after their removal, we present here the three immuno-inflammatory pathways which could be common to severe COVID, long COVID, and COVID vaccine side effects. We suggest that these three pathways could be responsible for long COVID and COVID side effects and explain in large part their strong similarities: the renin–angiotensin–aldosterone system, the kininogen–kinin–kallikrein (KKK) system (KKS), and the lectin complement pathway.

### 4.2. Dysregulation of the Renin–Angiotensin–Aldosterone System (ACE/Angiotensin II/AT1R Axis)

Various authors have summarized the impact of COVID on several organs and tissues, with a special focus on the significance of the renin–angiotensin–aldosterone system (RAAS) in the disease pathogenesis (Figure 2) [7,8,216]. The renin–angiotensin–aldosterone system (RAAS) is a key regulator of blood pressure and also contributes significantly to inflammatory processes [217]. Within this system, ACE2 exerts protective effects, including vasodilation, antioxidant activity, and anti-inflammatory actions. In contrast, angiotensin-converting enzyme (ACE) promotes vasoconstriction, oxidative stress, and pro-inflammatory responses. ACE2 mediates its protective effects by modulating levels of angiotensin II, a potent vasoactive peptide generated from angiotensin I through ACE activity. Due to the downregulation of ACE2 induced by viral binding and replication, the balance tilts in favor of the inflammation produced by angiotensin II, which is no longer degraded through the numerous mediators that result from it, especially in the lungs. This inflammation occurs through the interaction between angiotensin II and the angiotensin II type 1 receptor (AT1R) in the kidney and the vascular system [218,219]. Overall, over-activated AT1R results in vasoconstriction, hypertension, inflammation, oxidative stress, heart hypertrophy, tissue fibrosis (heart, lungs, kidneys, and liver), ageusia (loss of taste), anosmia (loss of smell), neurological dysfunctions, obesity and diabetes, and lesions in the skin [8,61].

This process leads to substantial cytokine production, including transforming growth factor-beta (TGF-β). Angiotensin II subsequently activates the nuclear factor kappa B (NF-κB) pathway, a key regulator of inflammation, which is also stimulated by other cytokines implicated in COVID-19, such as IL-1β, IL-6, TNF-α, IL-10, MCP-1, AT1, and platelet-derived growth factor beta (PDGF-B) [220,221,222].

In contrast, the ACE2/Ang-(1–7)/Mas receptor (MasR) axis represents the counter-regulatory arm of the RAAS, opposing the ACE/Ang II/AT1R axis (Figure 2). Angiotensin-(1–7) binding to MasR exerts vasodilatory, anti-proliferative, anti-fibrotic, anti-inflammatory, anti-angiogenic, and vasoprotective effects [223]. This axis suppresses inflammatory mediators such as NF-κB, IL-6, TNF-α, and IL-8 [198,224,225]. However, viral binding with ACE2 impairs this protective pathway, weakening its anti-inflammatory function.

Individuals with long COVID frequently exhibit a range of cardiovascular dysfunctions including myocardial inflammation, coronary microvascular impairment, arrhythmogenic disturbances, and blood pressure irregularities, conditions that have been mechanistically associated with disturbances in the RAAS [226]. Various authors have proposed to manage these complications with antagonists such as Telmisartan, which could re-establish cardiovascular homeostasis by blocking the actions of angiotensin II on AT1 receptors (AT1R) [216,226]. Also, other authors have proposed that pathophysiological mechanisms resulting from an unbalanced RAAS including a systemic hyperinflammatory state and coagulopathy, which are exacerbated by inflammation and oxidative stress, as well as fibrosis, may contribute to long COVID symptoms [227]. Accordingly, sustained stimulation of the angiotensin II type 1 receptor (AT1R) has been implicated in the pathophysiology of long-term sequelae following SARS-CoV-2 infection (long COVID), potentially contributing to adverse effects such as vasoconstriction, elevated blood pressure, chronic inflammation, oxidative stress, cardiac hypertrophy, fibrosis affecting multiple organs (including the heart, lungs, kidneys, and liver), as well as sensory impairments (such as anosmia and ageusia) and neurological dysfunctions [8,61]. One of the principal mechanisms proposed to underlie cardiovascular involvement in long COVID thus includes the interaction between the SARS-CoV-2 spike protein and the ACE2 receptor, subsequent ACE2 downregulation, and the cascade of tissue injury mediated by persistent immune activation and inflammatory processes [7,216].

Long COVID symptoms, whether arising after SARS-CoV-2 infection or following COVID vaccination, appear to involve intricate immune-inflammatory mechanisms, including dysregulation of the RAAS and an imbalance between bradykinin B1 and B2 receptor pathways [228]. The already mentioned ability of mRNA vaccine sequences encoding a recombinant spike protein to interact with the ACE2 binding site may permit for the vaccine-induced spike protein to subsequently destabilize RAAS in the whole body [207,208]. As previously mentioned in this review, free spike proteins found in circulation can reach every organ, even the brain by crossing the BBB, and induce inflammation in the whole body [170,171,201]. For these reasons, several other authors have assumed that the spike produced after vaccination could have a similar affinity for ACE2 as the virus, which could lead to similar undesirable effects as with COVID-19 and long COVID [213,214,229] and activate inflammation by imbalancing RAAS [230].

### 4.3. Activation of [Des-Arg^9^]-Bradykinin (ACE2/Bradykinin B1R/DABK Axis)

Infection with SARS-CoV-2 by depleting ACE2 increases the levels of des-Arg^9^-bradykinin (DABK) which is a known pulmonary inflammatory peptide factor. DABK, a bioactive bradykinin metabolite, is implicated in lung inflammation and injury via activation of bradykinin receptor 1 (B1R) on pulmonary endothelial cells. ACE2 degrades DABK, but this protective mechanism is disrupted by ACE2 downregulation following viral infection, resulting in uncontrolled activation of this pro-inflammatory pathway [231]. Notably, the interplay between the renin–angiotensin system, the kallikrein–kinin system, and the inflammatory and coagulation cascades is a key driver of COVID-19 pathophysiology.

Several studies have examined how SARS-CoV-2 infection affects cardiovascular function, emphasizing the potential disruption of the kinin–kallikrein system (KKS) as a contributing factor. This dysregulation, potentially triggered by viral interactions with host pathways, has been proposed as a therapeutic target, with implications for the development of future pharmacological strategies (Figure 3) [7,198,216,226]. Hyperactivation of pro-inflammatory pathways, including excessive cytokine release and dysregulated bradykinin signaling, has been associated with severe clinical manifestations and poor outcomes in COVID-19 patients.

A machine learning-driven proteomic investigation revealed 119 proteins significantly associated with long COVID among outpatient individuals (Bonferroni-adjusted *p* < 0.01). Further model optimization identified two minimal protein panels, comprising nine and five proteins, that demonstrated perfect discriminatory capacity between long COVID and control groups, with both models achieving an area under the curve (AUC) of 1.00 and F1 score of 1.00, indicating outstanding diagnostic accuracy. The identified proteins (CXCL5, AP3S2, MAX, PDLIM7, FRZB) reflected widespread organ and cell type expression [232]. Reduced ACE2 function is linked to activation of the des-Arg^9^ bradykinin (DABK)/bradykinin receptor B1 (BKB1R), potentially increasing neutrophil infiltration and the release of proinflammatory cytokines such as CXCL5 [231,233]. This latter mechanism is consistent with the elevated CXCL5 in long COVID outpatients found by this proteomic study [232]. Moreover, in a murine model of long COVID (K18-hACE2 mice), Sriramula S et al. (2023) found that elevated B1R expression may drive the long-lasting inflammatory response in the brain, driving chronic inflammation and ultimately cognitive and neuropsychiatric symptoms [234]. It is also noteworthy that several symptoms commonly reported in COVID-19, such as fatigue, nausea, gastrointestinal disturbances, and headaches, are similarly observed in disorders characterized by elevated bradykinin levels and increased vascular permeability, such as pulmonary angioedema [216,235,236].

Also, one hypothesis for the activation of KKS in long COVID is that a dormant SARS-CoV-2 infection of endothelial cells could lead to low-level expression and release of COVID-19 proteins on the cell surface, thereby perpetuating KKS and complement system activation [237]. It is also possible that this pathway remains overactivated by a prolonged AE2/ACE imbalance even in the absence of the spike protein.

As for RAAS activation, the ability of vaccine-induced spike proteins to bind with ACE2 receptors and fuse with them [207,208,213,214,229] together with the ability of mRNA and free spike proteins to reach several key organs via blood circulation make it possible for this spike protein, produced in a not fully controlled manner by COVID vaccines, to deregulate the ACE2/bradykinin B1R/DABK axis and produce DABK, subsequently resulting in inflammation similar to that observed in COVID and long COVID [170,171,201]. Interestingly, Cugno M et al. (2021) noted that after COVID vaccination, the physiopathological basis of the higher risk of urticaria/angioedema in subjects taking ACE inhibitors may be the increase in bradykinin, a vasoactive peptide that induces vascular permeability and that is catabolized by ACE [238].

Pharmacological agents modulating components of the RAAS and the KKS have been proposed as potential therapeutic strategies to mitigate long COVID manifestations and may apply as well for COVID vaccine side effects, in particular with C1 inhibitors (C1-INH) [226,236,239].

### 4.4. Lectin-Complement Pathway

A central role for the complement system, in particular lectin complement, in the pathogenesis of COVID-19 has emerged since the beginning of the pandemic outbreak (Figure 4) [7,198,199,200,216,240].

A number of pathophysiological characteristics including microangiopathy and myocarditis have been attributed to dysregulated complement activation due to SARS-CoV-2 infection. Furthermore, the analysis of >6500 proteins in 268 longitudinal samples revealed dysregulated activation of the complement system in individuals experiencing long COVID [241], an inflammatory pathway that we have described as being key in COVID as well as vaccine side effects, due to both N (nucleocapsid) and S (spike) protein implication [7]. Indeed, spike proteins have been shown to activate this pathway [200,242,243,244]. Authors found significantly lower levels of MBL and lectin pathway activity in the sera of patients experiencing brain fog as compared to recovered COVID-19 patients without brain fog [240]. Also, Baillie et al. observed that markers of the classical, alternative, and terminal complement pathway were markedly elevated in patients with long COVID [245].

After COVID vaccination, hypercoagulability like in VST has been attributed to a direct activation of complement pathways, promoting thrombin generation [184]. In COVID-19, as well as in long COVID and vaccine side effects, the spike protein can initiate inflammation in CNS through microglia and astrocytes and be linked to complement, bradykinin, and RAAS pathways [7,246,247]. Moreover, microglial activation and complement system activation share pathological mechanisms in various neurodegenerative disorders, including Alzheimer’s disease [248,249,250].

### 4.5. Common Cytokines Between COVID-19 Pathology, Long COVID, and COVID Vaccine Side Effects

The elevated levels of pro-inflammatory cytokines in the plasma aggravate systemic inflammation and lead to acute respiratory distress syndrome (ARDS), multi-organ failure, and death. Studies have shown increased levels of cytokines, particularly IL-1β, IL-6, TNF, and IL-10, in long COVID [162,251,252], which are the same or very common with key COVID cytokines [220]. It is important to note that systemic elevation of circulating cytokines may not always be detectable, as these molecules primarily exert paracrine and autocrine effects within local tissue environments rather than through widespread distribution in the bloodstream [253]. Studies have demonstrated that S1-induced neuroinflammation in microglia involves activation of the NF-κB and p38 MAPK pathways, leading to elevated production of pro-inflammatory mediators, including TNF-α, IL-6, IL-1β, and NO [212]. Sustained activation of immune-inflammatory pathways and cytokine production have been proposed as a key contributor to the prolonged clinical manifestations observed in individuals after SARS-CoV-2 infection [97,252].

In long COVID, studies have demonstrated significantly increased concentrations of Von Willebrand Factor, platelet factor 4, serum amyloid A, α-2antiplasmin E-selectin, and platelet endothelial cell adhesion molecule-1 in the soluble part of the blood [253]. The presence of microclotting, together with relatively high levels of these inflammatory molecules known to be key drivers of endothelial and clotting pathology, points to thrombotic endotheliitis as a key pathological process in long COVID like in COVID with microthrombosis and macrovascular thrombosis. This could help the research on long COVID therapies for the identification of patient cohorts by a battery of cytokine markers that have a persistent, low-level grade of inflammation in association with often two or more troubling symptoms [254].

Post-vaccination immune profiling has revealed significantly increased concentrations of several pro-inflammatory cytokines and chemokines (including IL-4, IL-6, CCL3 (*p* = 0.0012), CCL5, sCD40L, IL-8, and VEGF) in symptomatic individuals relative to controls, with the magnitude and duration of this inflammatory response showing a correlation with symptom persistence [255]. CCL3 has also been shown to be associated with long COVID patients as well as IL-6, TNF-α, and IL-17 [256].

### 4.6. The Importance of Oxidative Stress

The pivotal role of oxidative stress in the pathogenesis of COVID-19 and long COVID remains underappreciated and warrants further investigation to explore potential antioxidant and anti-inflammatory therapeutic strategies [7,257,258,259,260]. In addition, oxidative stress has emerged as a significant predictor of intensive care unit (ICU) admission, with particular emphasis on reduced plasma thiol concentrations (<154 µmol/L), demonstrating a sensitivity of 79.7%, specificity of 64.6%, positive predictive value of 58.8%, and negative predictive value of 78.9%. Other oxidative stress-related markers, such as advanced oxidation protein products (AOPPs), have also been associated with increased disease severity [259,261]. The main pathogenic processes of long COVID include a systemic hyperinflammatory state and coagulopathy, which are exacerbated by inflammation and oxidative stress [260]. Increased arterial stiffness, impaired endothelial function, and sustained oxidative stress have been implicated as potential mechanisms underlying persistent cardiac abnormalities observed in individuals with long COVID [262,263].

There is a vicious cycle between oxidative stress and inflammation, and long-COVID disease progression because of thrombosis and inflammation leads to the reactivation of reactive oxygen species (ROS). Inflammation is triggered by ROS, which also affects the endothelium, causes microthrombi and neuroinflammation, stimulates the production of autoantibodies, and interferes with the synthesis of neurotransmitters [227,260]. If oxidative stress and inflammation persist for months, even in the absence of spike proteins, and is not treated appropriately, it can contribute to the multitude of symptoms seen in long COVID [7,78]. Combined antioxidant and anti-inflammatory therapy have been proposed for treating chronic inflammation in long COVID patients [264,265]. An initial investigation reported that supplementation with Pycnogenol led to significant improvements in biomarkers of oxidative stress, alongside enhanced functional status and quality of life as reflected by elevated scores on the Karnofsky Performance Scale Index [266]. Also, it has been proposed that pharmacological activation of the nuclear factor erythroid 2-related factor 2 (NRF2) pathway may enhance the expression of antioxidant enzymes involved in glutathione biosynthesis, thereby augmenting intracellular antioxidant defenses and mitigating oxidative stress through the neutralization of reactive oxygen species [254]. PCSV and COVID vaccine side effects are also strongly associated with oxidative stress and glutathione (GSH), with GSH being responsible for maintaining mitochondrial function, antiviral defense, regulation of cellular proliferation, apoptosis, DNA synthesis, microtubular-related processes, and immune responses [267].

Oxidative stress in long COVID and vaccine side effects is driven by the three inflammatory pathways described in this review.

## 5. Discussion

Globally, ≈30–50% of individuals with COVID-19 experience long COVID symptoms, with regional variation. An analysis of 442 studies found that the prevalence was highest in South America (~51%), intermediate in Europe (~39%) and Asia (~35%), and lowest in North America (~30%) [268]. This comprehensive review encompassing global data sources identified studies investigating long COVID from six major continental regions. The distribution included 195 studies conducted in Europe, 126 in Asia, 61 in North America, 31 in South America, 9 in Africa, and 3 in Oceania. African estimates are variable and less reliable due to scarce data. Two comprehensive reviews conducted in Africa revealed that the pooled prevalence of long COVID was 41% (95% CI: 26–56%) [269] and 48.6% (95% CI: 37.4–59.8), with neuropsychiatric manifestations being the most commonly reported, exhibiting a cumulative incidence of 25% (95% CI: 21.1–30.4) [270]. Another comparative analysis (from 50 studies including almost 1.7 M individuals worldwide) also demonstrated geographical disparities in the prevalence of post-acute sequelae of SARS-CoV-2 infection (PASC), with estimated rates reaching 51% in Asian populations, 44% in European cohorts, and 31% in individuals from North America [271]. This meta-analysis identified fatigue, memory impairment, dyspnea, sleep disturbances, and arthralgia as the most frequently reported symptoms among individuals with long COVID. Their respective pooled prevalence estimates were fatigue at 23% (95% CI: 17–30%), memory difficulties at 14% (95% CI: 10–19%), shortness of breath at 13% (95% CI: 11–15%), sleep-related issues at 11% (95% CI: 5–23%), and joint pain at 10% (95% CI: 4–22%). Also, globally, the prevalence of long COVID is generally higher among hospitalized individuals (≈44–54%) vs. non-hospitalized (≈34%) [268].

As a reminder, as of 31 January 2022, a total of 349,641,119 (4486 per 100,000) confirmed COVID-19 cases had been documented globally based on the World Health Organization (WHO) COVID-19 Dashboard [38]. Among all regions, Europe and the Americas reported the highest cumulative incidence at 14,040 per 100,000 and 12,513 per 100,000, respectively. The global cumulative number of COVID-19-related deaths reached 5,592,266, resulting in a global mortality rate of 71.75 deaths per 100,000 population. The Americas exhibited the highest cumulative death rate (241.63 per 100,000), with Europe ranking second (186.75 per 100,000) and South-East Asia (36.16 per 100,000) and Africa (14.49 per 100,000) far behind.

Regarding post-COVID-19 vaccine syndromes (PCVS), Ogar CK et al. (2023), with VigiBase data (WHO’s global database of infrastructure for post-introduction pharmacovigilance/ICSRs) and using a mixed-methods comparative analysis, found ~180 adverse effects (AEs) per million doses for Africa, ~3324 AEs per million doses for the Americas, ~4,549 AEs per million doses for Europe, ~21 AEs per million doses for Southeast Asia, and ~272 AEs per million doses for the Western Pacific, with a global average of ~1278 AEs per million doses [272]. Geographic variation in adverse event reporting appears substantial but likely reflects differences in surveillance capacity and reporting practices rather than true biological differences in vaccine safety. The proportion of serious adverse events (SAEs) among all reported adverse events (AEs) was 12.2% in reports originating from Africa, whereas it reached 27.1% in data from other continents. Fatal outcomes were cited as the underlying cause of seriousness in 10.1% of SAEs reported from Africa, compared to 9.6% in the rest of the world. At the time of reporting, 61.4% of adverse events (AEs) submitted from the African region were documented as resolved, a proportion comparable to the 58.2% resolution rate observed in reports originating from other global regions.

Accurate quantification of severe adverse events remains challenging due to the absence of consensus and underreporting across surveillance systems. Notably, the U.S. Vaccine Adverse Event Reporting System (VAERS) has generated safety signals that, in some cases, include reports of associated mortality. However, these figures are likely to be substantially underestimated, as it is widely acknowledged that spontaneous reporting systems capture only a fraction, approximately 10%, of actual adverse events [273].

Given the widespread manifestations of long COVID, particularly its cardiovascular and neurological sequelae, elucidating the underlying pathophysiological mechanisms is essential for the development of targeted interventions aimed at reducing long-term health consequences and mitigating the risk of chronic diseases. Such efforts are critical for stratifying individuals at higher risk of mortality and for predicting those susceptible to persistent cardiovascular and neurological complications, thereby informing more effective therapeutic approaches. Long COVID is particularly worrying when it develops to “chronic COVID”, i.e., where symptoms remain for more than 12 weeks [6,8]. The symptoms of long COVID consist of damage to various organs and systems such as the respiratory, cardiovascular, neurological, endocrine, urinary, and immune systems but seem to particularly affect the CNS, vascular endothelium, and cardiovascular system. That is why the associated symptoms are as diverse as fatigue, headache, smell loss, cognitive and attention impairments, sleep disturbances, VST, TM, Guillain–Barré syndrome and optic neuritis [31], dyspnea, persistent cough, cardiac abnormalities such as myocarditis, microvascular angina, cardiac arrhythmias, pressure abnormalities, and thrombosis (Table 1) [16,17,19].

Importantly, most long COVID symptoms, if not all, are also observed as mild or severe side effects of mRNA or adenoviral DNA COVID vaccines affecting respiratory, cardiovascular, neurological, endocrine, urinary, and immune systems [6,7,22,24,29,40,41,42,43,118,149,274,275,276,277]. In this review, we emphasize the elements which suggest that the pathophysiology of long COVID and COVID vaccine side effects share some similarities, and both have significant connections with the renin–angiotensin–aldosterone system (RAAS), the kininogen–kinin–kallikrein system (KKS), and the lectin-complement immune-inflammatory pathway. Long COVID, especially in the persistence of the virus, and vaccines have in common the presence or the chronic consequences of the toxic spike protein, which can induce these three pathways in the whole body and organs, including the vascular endothelium, heart, and brain. Indeed, spike proteins with SARS-CoV-2 as well as free spike proteins and mRNA from COVID vaccines can travel in the circulation, reach a great number of organs, and also pass the BBB to reach all brain areas, where they can produce inflammation through the interaction of spike proteins with their ACE2 receptor and activation of the RAAS and bradykinin pathways [7,216,227,230]. Spike proteins can also activate the lectin-complement pathway by binding to MBL (mannan-binding lectin)-MASP-2 (mannan-binding lectin serine protease 2) and also induce chronic inflammation [7,242].

Furthermore, a study has shown that receiving two doses of the COVID-19 vaccine could increase by 2.32 the risk of having long COVID, which indicates that vaccine-induced spike proteins, by activating RAAS, bradykinin, and lectin-complement pathways, could lead to long COVID symptoms or long COVID aggravation [56]. Given the similarities in symptomatology and possible pathophysiological mechanisms (as outlined in Table 1), it is plausible that a fraction of the estimated 400 million global long COVID cases may also be associated with COVID-19 vaccination-related sequelae. This is not an affirmation, but we think that this question deserves to be addressed by further clinical studies and analyses as well as laboratory investigations. While COVID-19 vaccines were considered a pragmatic solution to mitigate the pandemic, it remains scientifically pertinent to question whether administering them to vulnerable individuals with comorbidities, such as conditions frequently associated with chronic inflammation [278,279], was optimal given that these vaccines may activate similar immuno-inflammatory, oxidative stress, immune dysregulation, and prothrombotic pathways as the SARS-CoV-2 virus itself. As detailed in this review, the “intact structure” of the spike protein produced by the vaccines permits this protein to bind to the same ACE2 receptors and may activate the same biochemical pathways as the spike proteins from SARS-CoV-2. Also, during the pandemic, health authorities and many physicians and researchers pushed for COVID vaccination even in patients with a prior history of acute myocarditis and tried to reassure them based on the fact that it was not associated with a risk of myocarditis recurrence or serious side effects [135]. Regarding the large literature on myocarditis and severe cardiac issues induced by COVID vaccines, this is really debatable. Another example could be the increase in BBB permeability through tight junction alterations observed in inflammatory conditions such as diabetes, which should be considered in the context of vaccination strategies [271].

We want to point out here that, despite a lot of similarities in symptoms and pathophysiology observed between long COVID and vaccine side effects (PACVS), we do not consider these two conditions to be the same. Also, we advocate that further studies on the three biochemical pathways, which we propose to be the most relevant, should be conducted, including studies using treatment blocking these pathways.

Indeed, regarding the activation of complement pathways, for example, this implication has been well established in the development of severe COVID disease and mortality [7,198,199,200,280]. Regarding this pathway in long COVID, the analysis of >6500 proteins in 268 longitudinal samples revealed dysregulated activation of the complement system in individuals as published in the journal *Science* [241]. In this work, sustained complement system activation initiated during the acute infection phase was still present after a 12-month follow-up period, and normalization of complement protein concentrations was observed exclusively in patients who experienced clinical resolution of long COVID [241]. Baillie et al. (2024) also observed that markers of complement pathways were markedly elevated in patients with long COVID (*n* = 166 vs. controls, *n* = 79) [245], and they proposed “a novel set of biomarkers that could guide the diagnosis and treatment of long COVID” and that “currently available inhibitors of complement activation could be used to treat long COVID”. In a prospective cohort study involving 147 prospectively followed COVID-19 patients and 216 healthy controls, MASP-2 levels exhibited significant positive correlations with the terminal complement complex (ρ = 0.3596, *p* < 0.0001), the lectin pathway recognition molecules ficolin-2 (ρ = 0.2906, *p* = 0.0004) and ficolin-3 (ρ = 0.3952, *p* < 0.0001), as well as with C-reactive protein levels (ρ = 0.3292, *p* = 0.0002) [281]. Furthermore, the comparative analysis revealed no statistically significant variation in MASP-2 serum levels between individuals recovered from COVID-19 (*n* = 347) and uninfected control subjects. Also, in lung tissue from patients with non-resolvable COVID-19 (long COVID), elevated mRNA expression of MASP-2 and associated complement factors (C4a/C4b) suggests sustained lectin and classical pathway activity contributing to progressive fibrosis [282]. Importantly, an investigation revealed a marked reduction in serum mannose-binding lectin (MBL) concentrations and attenuated lectin complement pathway activity among brain fog-positive individuals, while the functional integrity of the classical and alternative complement cascades remained unaffected (patients from Neuro-Long-COVID ambulatory service of the University Hospital of “Cattinara”—Trieste, Italy—between 1 November 2021 and 1 March 2022) [240]. Cognitive dysfunction observed in long COVID, commonly referred to as “brain fog”, may represent one of the clinical manifestations linked to mannose-binding lectin (MBL) deficiency. Additionally, in a single-center study cohort of 74 hospitalized COVID patients, lectin and alternative pathways were strongly activated, and antigen levels of complement components (MBL, C4, C3, Factor B, C5) were low in the high-activation cluster, indicating consumptive depletion related to severe illness [242]. In consequence, these correlations highly suggest the need to follow complement markers and test the efficacy of complement inhibitors in COVID-19, long COVID, and for patients suffering from COVID vaccine side effects, in order to prove or invalidate the relevancy of this hypothesis. Altogether, these results do not provide causal proof of the implication of lectin complement in the pathophysiology of long COVID, but they justify the interest in further studies, as spike proteins have been shown to activate MASP-2 and the lectin route [200,242,243,244].

Of note, regarding PACS and PCVS, even if long COVID symptoms may often be more common than COVID vaccine side effects, the latter must not be neglected, and both have to be studied more thoroughly in order to find the best treatments as soon as possible. Clinicians and scientists are increasingly encountering patients reporting persistent symptoms post-SARS-CoV-2 infection or vaccination which need attention and treatments [6]. Indeed, in the context of post-acute COVID-19 syndrome (PACS), the “serious implications for individuals and society have been missing from public communication and pandemic policy” [283]. Patients also report “encountering medical professionals who dismissed their experience, leading to lengthy diagnostic odysseys and lack of treatment options for long COVID” [284]. This experience, termed “medical gaslighting,” must be avoided for both PACS and vaccine-related syndromes like PCVS. Notably, “long COVID” was a term coined by patients themselves, not clinicians [285]. Turner et al. emphasize that “there is hesitancy among patients and researchers to acknowledge and openly discuss vaccine injury, due to fear of being labeled ‘anti-vax’.” [39]. They argue that affected individuals must receive care “without fear of being stigmatized,” and that such injuries “should be researched like any other disease” to support safer vaccine development and identify at-risk populations.

Myocarditis, for example, has to be seriously controlled after both long COVID and vaccine reaction. Myocarditis has generally shown to be increased after COVID infection: in a cohort of over 20 million individuals followed for 9.5 months, COVID-19 survivors showed a myocarditis incidence of 21 per 100,000 (95% CI: 0.13–0.42), compared to 9 per 100,000 (95% CI: 0.07–0.12) in uninfected controls, indicating a significantly higher risk post-infection [286]. Also, myocarditis induced by COVID injection has been often described to be higher than after COVID vaccination, but both conditions need to be carefully addressed, as shown by several studies all over the world.

In Israël, a population-based study found that COVID-19 vaccination was linked to a modestly increased myocarditis risk, with a risk ratio (RR) of 3.24 (95% CI: 1.55–12.44), corresponding to 2.7 excess cases per 100,000 individuals (95% CI: 1.0–4.6) [287]. In comparison, SARS-CoV-2 infection conferred a markedly higher risk, with an RR of 18.28 (95% CI: 3.95–25.12) and an absolute increase of 11.0 myocarditis cases per 100,000 persons (two groups of 884,828 persons were compared). Using the same Health Care Organization (HCO) database with a more heterogeneous population (median age 44 vs. 38 years) and employing manual case review on complete clinical data, Witberg et al. (2021) identified 54 myocarditis cases within 42 days post-COVID-19 vaccination among over 2.5 million individuals aged ≥16 [288]. The overall incidence was 2.13 per 100,000 (95% CI: 1.56–2.70), peaking at 10.69 per 100,000 (95% CI: 6.93–14.46) in males aged 16–29. About 76% of cases were mild and 22% moderate. In another study in Israel on 5,442,696 vaccinated subjects, the incidence of myocarditis after a second dose of BNT162b2 was 15.1 per 100,000 in 16- to 19-year-old males [289].

Others data from EMA also found a significant number of excess cases of myocarditis after the second dose of mRNA vaccine in young vaccinees compared to unexposed: 2,6 (French data) and 5.7 (Nordic data) per 100,000 for Comirnaty and 13 and 19 per 100,000, respectively, for Spikevax [290]. Also, a Nordic cohort study of 23.1 million individuals found increased myocarditis risk within 28 days after mRNA COVID-19 vaccination, especially after the second dose and among males aged 16–24. The risk was higher with mRNA-1273 (9–28 excess cases per 100,000) than with BNT162b2 (4–7 per 100,000) [291].

This was the case in Canada as well, where significant myocarditis occurred after COVID vaccination, especially with shorter intervals between injections: out of 19.7 million mRNA vaccine doses, 297 cases of myocarditis/pericarditis were reported (15 per 100,000), mostly in males (76.8%) with a median age of 24. Most cases (69.7%) occurred after the second dose. The highest incidence was in males aged 18–24 after a second dose of mRNA-1273 (29.95 per 100,000), much higher than after BNT162b2 (5.92 per 100,000) [134]. In Hong Kong, Chua GT et al. (2022) reported 33 adolescent myocarditis/pericarditis cases after BNT162b2 vaccination, mostly in males (87.9%, median age 15.25) [292]. Most occurred after the second dose (81.8%) and were mild. The overall incidence was 18.52 per 100,000, rising to 37.32 per 100,000 among males after the second dose. In conclusion, the sequelae of COVID infection are often more common than those of COVID vaccine cases, but both patients deserve attention and treatments, beyond the issue of the number of events, which still remains to be specified.

To conclude on this key topic, we would like to share very interesting thoughts from several high-ranked researchers on COVID, which was published in *Science* by Couzin-Frankel J and Vogel G (2022) titled “Vaccines may cause rare, long COVID-like symptoms.”: “Now, a small number of other researchers worldwide is beginning to study whether the biology of long COVID, itself still poorly understood, overlaps with the mysterious mechanisms driving certain postvaccine side effects.” [293]. William Murphy, an immunologist at the University of California, Davis, in November 2021 in *The New England Journal of Medicine*, proposed that an autoimmune mechanism triggered by the SARS-CoV-2 spike protein might explain both long Covid symptoms and some rare vaccine side effects, and he called for more basic research to probe possible connections [294].

Resia Pretorius (a physiologist at Stellenbosch University in South Africa) adds that “she and her colleagues have also seen patients, fewer than 20, she estimates, with chronic problems following vaccination”. She says, “these include long COVID–like symptoms such as brain fog as well as other clotting concerns such as deep vein thrombosis. The cause of the very rare but severe clotting after the AstraZeneca and Johnson & Johnson vaccines remains unknown, but Pretorius suspects all COVID-19 vaccines might also sometimes trigger subtler clotting issues.” Indeed, the ChAdOx1 nCoV-19 vaccine has been associated with a pooled venous thrombosis incidence of 28 per 100,000 administered doses (95% CI: 12–52; I^2^ = 100%) [295]. In particular, the incidence of cerebral venous thrombosis following vaccination was estimated at 23 per 100,000 person-years, substantially higher than the pre-pandemic background rate of 0.9 per 100,000 person-years.

Avindra Nath, clinical director at the National Institute of Neurological Disorders and Stroke (NINDS), submitted a case series of 23 people to a third publication, and says “his group has submitted an amendment to a long COVID protocol to include patients with postvaccine side effects.” At Yale School of Medicine, Akiko Iwasaki is planning to collaborate with Avindra Nath and look at any potential link between long Covid and post-vaccine effects. Also, Susan Cheng (director of public health research in the Smidt Heart Institute at Cedars-Sinai) and her col-leagues “are planning a case series that includes sophisticated imaging and diagnostic tests from a mix of long COVID patients and those with postvaccine side effects.” as “she finds the overlap between their symptoms and those of long COVID compelling.

All these researchers say “We shouldn’t be averse to adverse events”. “Researchers exploring postvaccine side effects all emphasize that the risk of complications from SARS-CoV-2 infection far outweighs that of any vaccine side effects”, says Harald Prüss (neurologist at the German Center for Neurodegenerative Diseases (DZNE) and the Charité University Hospital in Berlin). “But understanding the cause of postvaccine symptoms, and whether early treatment can help prevent long-term problems, could be crucial for designing even safer and more effective vaccines, as well as potentially providing clues to the biology of Long Covid”, says William Murphy (immunologist at the University of California, Davis).

We are aligned with these statements and strategy, and we also want to remind that it is of particular importance to use early treatments and anti-inflammatory and anticoagulants medicines, as advised by a many other scientists quoted in this review, in order to better treat COVID and also to manage long COVID symptoms as soon as possible, before reaching the point of severe complications, which can ultimately lead to more complicated chronic diseases. We propose in this review that the blockade of RAAS, KKS, and lectin-complement pathways should be a strategy to improve both long COVID and vaccine side effects. For this purpose, there is still an urgent need for more clinical studies of well-defined groups of long COVID and COVID vaccine patients for finding therapeutic solutions faster, in particular targeting these three immuno-inflammatory pathways [296].

## Figures and Tables

**Figure 1 ijms-26-07879-f001:**
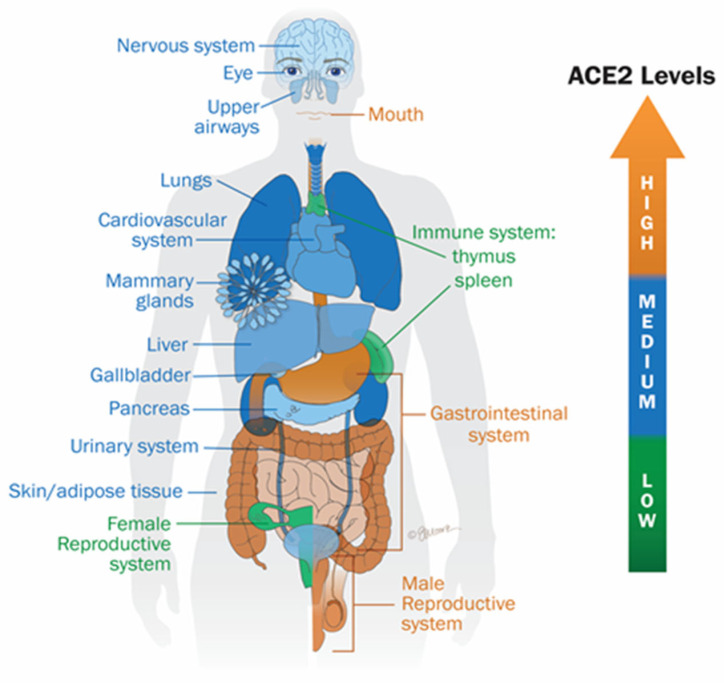
Distribution of ACE2 receptor expression: The color gradient, ranging from orange (high expression) to green (low expression), illustrates the relative levels of ACE2 expression across various tissues and body fluids. Notably, the highest expression levels are observed in the oral cavity, gastrointestinal tract, and male reproductive organs. Adapted from Lesgards et al. (2023) [7].

**Figure 2 ijms-26-07879-f002:**
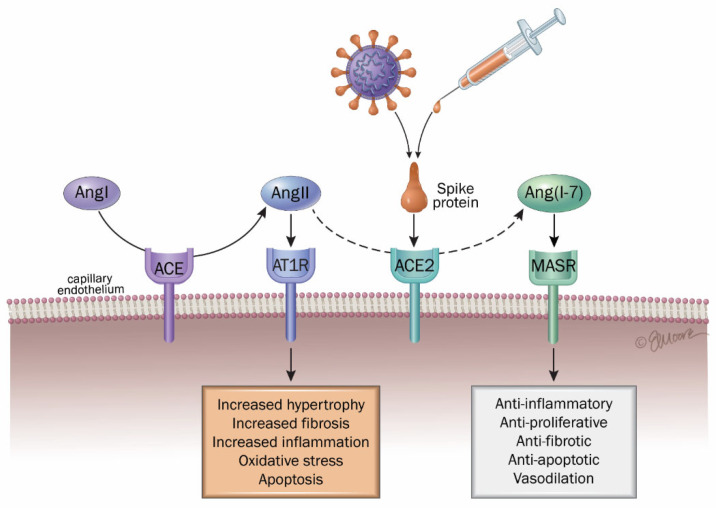
Schematic overview of RAAS perturbation by spike protein. Adapted from Lesgards et al. (2023) [7]. Angiotensinogen, a precursor protein synthesized and secreted by the liv-er, is cleaved by renin (a proteolytic enzyme released by the kidneys) into angiotensin I (Ang I). Ang I is subsequently converted to angiotensin II (Ang II) through the catalyt-ic activity of angiotensin-converting enzyme (ACE). Ang II exerts its physiological effects primarily by binding to angiotensin II type 1 (AT1) and type 2 (AT2) receptors. Activation of the AT1 receptor (AT1R) by Ang II promotes vasoconstriction, cellular proliferation, fibrosis, and pro-inflammatory responses. ACE2 converts Ang-I and Ang-II to angiotensin (1–7). Ang (1–7) binds to the MAS receptor (MASR) to promote vasodilation, vascular protection, anti-fibrosis, anti-proliferation, anti-inflammation, and anti-angiogenesis actions.

**Figure 3 ijms-26-07879-f003:**
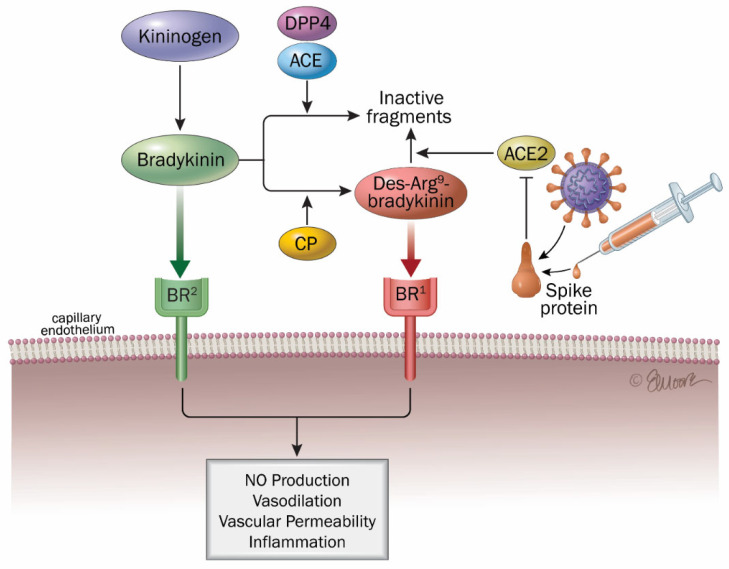
Schematic overview of the kininogen–kinin–kallikrein (KKK) system (KKS) perturbation by spike protein. A. Angiotensin-converting enzyme (ACE) degrades bradykinin, a vasodilatory peptide that primarily signals through bradykinin B2 receptors (B2R). Bradykinin can also be processed by kininase I to form des-Arg^9^-bradykinin (DABK), which exerts vasoconstrictive and pro-inflammatory effects via bradykinin B1 receptors (B1R). B. SARS-CoV-2 infection disrupts the kallikrein–kinin system (KKS), favoring the accumulation of DABK and enhancing pro-inflammatory signaling. Adapted from Lesgards et al. (2023) [7].

**Figure 4 ijms-26-07879-f004:**
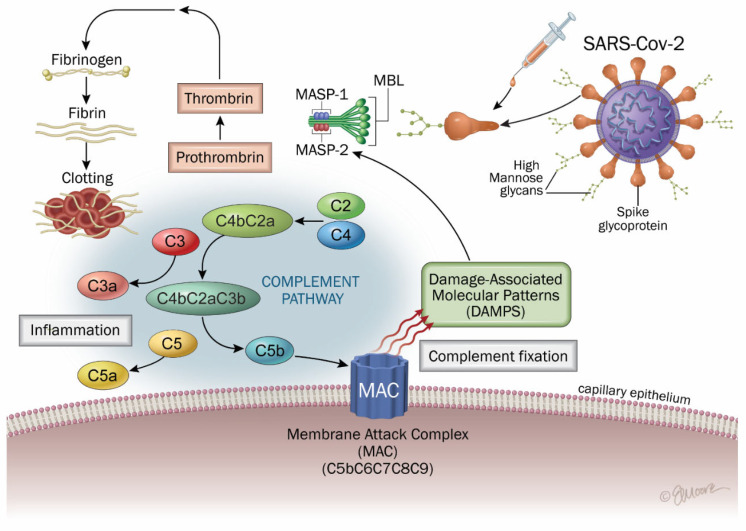
Lectin pathway activation by SARS-CoV-2 and spike protein from mRNA vaccines. The lectin complement pathway is activated when mannose-binding lectin (MBL)/mannan-binding lectin serine protease 2 (MASP-2) binds to pathogen surfaces. This complex cleaves complement components C4 and C2, generating the C3 convertase (C4b2a). The classical pathway is triggered by binding of the C1 complex to immunoglobulins or endogenous ligands, also resulting in C4 and C2 cleavage and formation of the C3 convertase (C4b2a). C3 convertase then cleaves C3 into C3a, an anaphylatoxin, and C3b, promoting the assembly of the C5 convertase (C4b2a3b or C3bBb3b). This leads to cleavage of C5 into C5a and C5b, with C5b initiating the terminal complement cascade, culminating in membrane attack complex (MAC, C5b-9) formation and pathogen lysis. Adapted from Lesgards et al. (2023) [7].

**Table 1 ijms-26-07879-t001:** Summary of symptoms associated with long COVID and post-COVID-19 vaccination syndrome (PCVS) and their related systems.

Symptoms	Systems	Long COVID (References)	Post-COVID-19 Vaccination Syndrome (PCVS) (References)
Fatigue	Nervous system	[6,13,15]	[20,21]
Headache		[6]	[20,22]
Attention deficit		[19,23]	[24]
Cognitive impairment		[25,26]	[6]
Muscle pain		[17,27]	[6]
Anxiety		[16,28]	[29]
Depression		[16,28]	[30]
Transverse myelitis		[6]	[21,22]
Guillain–Barré syndrome		[6]	[21,22]
Optic neuritis		[31]	[21,22]
Cerebral venous sinus thrombosis		[6]	[21,22]
Anosmia		[17]	[6]
Ageusia		[17]	[6]
Myocarditis	Cardiovascular system	[17,32]	[7,33]
Pericarditis		[17,34]	[35,36]
Heart failure		[27]	[37,38]
Thrombosis		[39]	[7,40]
Ischemic heart disease		[10]	[41]
Dysrhythmias		[16,27]	[42]
Blood pressure disorder		[27]	[43]
Ischemic stroke		[17,27]	[44]
Cardiac arrest		[10]	[41]
Dyspnea	Respiratory system	[8,17,19]	[8]
Persistent cough		[27]	[6]
Chest pain		[17]	[41]
Pneumonia		[6]	[7]
Activation of innate immune cells	Immune system	[45,46]	[7,47]
Inflammation		[7]	[7]
Mast cell activation syndrome		[45,46]	[43]
Macrophage activation syndrome		[48,49]	[7,50]

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
