# Peer review of "Do Long COVID and COVID Vaccine Side Effects Share Pathophysiological Picture and Biochemical Pathways?"

_ijms, 2025, doi:10.3390/ijms26167879_

Round 1

Reviewer 1 Report

Comments and Suggestions for Authors

Dear authors, your review about Long COVID and COVID vaccine side effects is very interesting and thorough. The strong point is the structure of the review since from the definition of Long COVID with pathological manifestations, the predictive and intervention factors, we then move to a more specific focus on some aspects of the damage to certain organs given by the infection and the effects of the vaccine.
From my point of view, I would anticipate chapter 3 on biochemical aspects before point 2.
I would also introduce some more or at least clearer data on the geographical spread of cases of long-term COVID and COVID vaccine side effects in America, Europe, Asia, etc. 

Author Response

Dear reviewer, thanks for attention to the reviewing of our work and your remarks.

Comment 1: Regarding the geographical spread of cases of long-term COVID and COVID vaccine side effects in America, Europe, Asia, etc., we added a special part on that interesting point at the beginning of the discussion, at page 49.

Comment 2: You proposed that we “anticipate chapter 3 on biochemical aspects before point 2.”

Answer 2: Regarding your point on anticipating chapter 3 on biochemical aspects before point 2, we also had this reflection, but we would prefer to leave these parts in this order. In fact, the observation of similarities between long COVID symptoms and pathophysiology with COVID vaccine side effects, was the starting point of our will to do this review. Indeed we are researchers and also physicians and pharmacists who dealt with patients during COVID and today with long COVID. And that led us to investigate on understanding the associated biochemical pathways in order to propose tracks for treatments. We also think that for a majority of physicians, researchers as well as for the public, long COVID represents a multitude of symptoms, which is true, and we wanted to start from that point to lead them towards biochemistry. Indeed, we think modestly but with conviction, that many therapies which have been proposed during COVID without directly trying to assess precisely the precise pathways, which lead to COVID severe cases and deaths. We hope you understand our logic.

Best regards,

Reviewer 2 Report

Comments and Suggestions for Authors

Author Response

Dear reviewer, thanks for attention to the reviewing of our work and your remarks.

Comment 1: “For publication in a high-impact journal such as IJMS, the manuscript requires major revisions to address scientific rigor, tone, and balance. I think the two issues are not the same. I suggest the title could be rephrased as follows: " Do Long COVID and COVID Vaccine Side Effects Share Pathophysiological Pictures and Biochemical Pathways?"”

Answer 1: We have tried our best, in this revised article, in order to improve the rigor, tone and balance of our review.

We agree to change the title as such: “Do Long COVID and COVID Vaccine Side Effects Share Pathophysiological Picture and Biochemical Pathways?”

Comment 2: “Lack of the term of Introduction in Section 1. Such mistakes limit the scientific merits. “

Answer 2: Thanks for correcting this error: we had an introduction of the review.

Comment 3: “The manuscript frequently implies causality (e.g., spike protein causing myocarditis or neurological damage post-vaccination) based on correlational or anecdotal evidence. Many of the cited studies report associations, not mechanism or causation.”

Answer 3: We tried to improve our arguments and moderate on causality, but highlight and strengthen the links between long COVID and COVID vaccine side effects with the need of further studies on this topic, all along the manuscript.

We had a special paragraph on this point p54:

“We want to point out here that, despite a lot of similarities in symptoms and pathophysiology observed between long COVID and vaccine side effects (PACVS), we do not consider these 2 conditions to be the same. Also we advocate that further studies on the 3 biochemical pathways, which we propose to be the most relevant, should be conducted, including studies using treatment blocking these pathways…”

End of the paragraph is: “Altogether these results do not bring a causal proof of the implication of lectin complement in the pathophysiology of long COVID, but it justifies the interest for further studies as spike protein has been shown to activate MASP-2 and lectin route (191) (233-235).”

Comment 4: “Statements such as “the spike protein produced by these COVID vaccines is able to… destabilize RAAS in all the organism” are overgeneralized and speculative.”  

Answer 4: We have suppressed all the sentences which sounded like statements and replaced them with moderation “may”, “could”, “possible” etc.

Comment 5: “I caution authors to be aware of conflation of Vaccine Side Effects and Long COVID. “

Answer 5: We have added statements that it is not the same disease but also tried to defend our point on the similarities. In the paragraph on comment 4 “We want to point out here that, despite a lot of similarities in symptoms and pathophysiology observed between long COVID and vaccine side effects (PACVS), we do not consider these 2 conditions to be the same” but also with moderations along the text.

Meanwhile we try to defend our hypothesis of close pathologies and added a special paragraph in the end of the discussion (bottom of page 58) published in Science and they make close remarks to our hypothesis and include patients suffering vaccine side effects with long COVID patients (Couzin-Frankel J and Vogel G (2022), “Vaccines may cause rare, long COVID-like symptoms.”). 

Extracts: “Now, a small number of other researchers worldwide is beginning to study whether the biology of long COVID, itself still poorly understood, overlaps with the mysterious mechanisms driving certain postvaccine side effects.”(291). Murphy, an immunologist at the University of California, Davis. In November 2021 in The New England Journal of Medicine, proposed that an autoimmune mechanism triggered by the SARS-CoV-2 spike protein might explain both Long Covid symptoms and some rare vaccine side effects, and he called for more basic research to probe possible connections (292).

“Avindra Nath, clinical (director at the National Institute of Neurological Disorders and Stroke (NINDS) submitted a case series of 23 people to a third publication, and says « his group has submitted an amendment to a long COVID protocol to include patients with postvaccine side effects. » At Yale School of Medicine, Akiko Iwasaki is planning to collaborate with Avindra Nath and look at any potential link between Long Covid and postvaccine effects.”

On your other point: “Serious side effects after vaccination are rare and tightly monitored. Please acknowledge differences in incidence, scale, severity, as well as risk rates (e.g., vaccine myocarditis in young males vs post-COVID myocarditis in all ages).”

We underlined that without minimizing the PCVS with studies comparing teh events. We addressed your point on myocartis by a special party in the discussion p 56:

Extracts: “Also, myocarditis induced by COVID injection has been often described to be higher than after COVID vaccination but both conditions require to be carefully addressed as shown by several studies all over the world.

In Israël, a population-based study found that COVID-19 vaccination was linked to a modestly increased myocarditis risk, with a risk ratio (RR) of 3.24 (95% CI: 1.55–12.44), corresponding to 2.7 excess cases per 100,000 individuals (95% CI: 1.0–4.6)  (285). In comparison, SARS-CoV-2 infection conferred a markedly higher risk, with an RR of 18.28 (95% CI: 3.95–25.12) and an absolute increase of 11.0 myocarditis cases per 100,000 persons (2 groups of 884,828 persons were compared).”

This part ends like this “In conclusion, meanwhile the sequelae of COVID infection are often more common than those of COVID vaccine cases, both patients deserve attention and treatments, beyond the issue of the number of events, which still remains to be specified.”

Comment 6: Please avoid alarmist language. Phrases like “lack of seriousness of the companies,” “proofs have been provided,” and “uncontrolled production of spike protein” are subjective and speculative. 

The manuscript sometimes borders on advocacy rather than scientific neutrality. Please use neutral, scientific language throughout and avoid speculative accusations unless they are supported by rigorous data and peer-reviewed investigations reported by news.

We removed what could be seen as alarmist: “lack of seriousness of the companies,” “proofs have been provided,” and “uncontrolled production of spike protein”. As said, we improved the tone and language.

All the paragraph was deleted even if the delay for preparing this vaccine was very short (p21):

Due to a lack of seriousness of the companies involved in COVID vaccination, observation period of the approval and post-marketing studies did not take this long period of time into account. Proofs have been provided that vaccination is the cause of the disease thanks to the detection of mRNA and/or spike proteins from the COVID-19 vaccine. 

Below for ex, for moderation “which suggests an implication of vaccine-induced spike proteins rather than the virus itself (117). More studies reporting the presence of both N and S proteins are needed in this matter in order to understand the consequences of long COVID vs. vaccine side effects. “

Or “These findings suggest that vaccination could, in some cases, elicit symptoms similar to those seen in COVID-19 and long COVID, and may theoretically contribute to the development or exacerbation of inflammatory conditions over the medium to long term, including cardiovascular, neurological, oncological, and autoimmune disorders. “ p35.

Comment 7: The manuscript sometimes borders on advocacy rather than scientific neutrality. Please use neutral, scientific language throughout and avoid speculative accusations unless they are supported by rigorous data and peer-reviewed investigations reported by news.

Answer 7: We reformulated and concluded like this (p18): “Consequently, this would explain the persistence/recurrence of cardiovascular and neurological complications following SARS-CoV-2 infection representing part of the symptoms of long COVID but this hypothesis is considered as very unlikely and would require an analysis of human genome in patients, which has not been done yet.”

Comment 8: “The article often combines retrospective data, case series, autopsy reports, and in vitro experiments without discussing strength of evidence, confounding, or controls. Please add a methodological tables or figures summarizing the strength and type of evidence behind each claim to provide Distinctions Between Observational and Experimental Studies”

Answer 8: We have detailed more studies (with type controls) in the text and tried to better explain our claims and the similarities between long COVID and vaccines side effects, but without adding tables. 

We agree that our hypotheses require further studies and we thank you for your remarks which permit us to improve our manuscripts and thoughts on this subject.

Minor concerns:

Comment 1: “Please clearly distinguish “post-acute COVID-19 syndrome (PACS),” “long COVID,” and “post-COVID-19 vaccination syndrome (PCVS)” based on Terminology Consistency. “

Answer 1: We tried to precise at the beginning of the part dedicate and we end by this :”In this review we are using the general term post-COVID-19 vaccination syndrome (PCVS) as we focus on post-acute COVID-19 vaccination syndrome (PACVS) i.e. chronic manifestations only. “

Comment 2: “Please provide a list of abbreviations and define all acronyms on first use (e.g., RAAS, KKS, PCVS)”

Answer 2: We have verified that all repeated terms in the text have been defined and introduced 1 or 2 times: as a whole together with abbreviation.

Best regards,

Round 2

Reviewer 2 Report

Comments and Suggestions for Authors

I agree with the aurthors’ revision and endorse the publication.